behaviour

fast saccades, numerosity perception, attention, subcortex

**Author for correspondence:**
David Burr
e-mail: davidcharles.burr@unifi.it

# Fast saccadic eye-movements in humans suggest that numerosity perception is automatic and direct

Elisa Castaldi[1,2], David Burr[2,3], Marco Turi[4] and Paola Binda[1]

[1]Department of Translational Research and New Technologies in Medicine and Surgery, University of Pisa, Pisa, Italy
[2]Department of Neuroscience, Psychology, Pharmacology and Child Health, University of Florence, Florence, Italy
[3]Institute of Neuroscience, National Research Council, Pisa, Italy
[4]Stella Maris Mediterraneo Foundation, Chiaromonte, Italy

EC, 0000-0003-0327-6697; DB, 0000-0003-1541-8832; MT, 0000-0002-4495-0804; PB, 0000-0002-7200-353X

Fast saccades are rapid automatic oculomotor responses to salient and ecologically important visual stimuli such as animals and faces. Discriminating the number of friends, foe, or prey may also have an evolutionary advantage. In this study, participants were asked to saccade rapidly towards the more numerous of two arrays. Participants could discriminate numerosities with high accuracy and great speed, as fast as 190 ms. Intermediate numerosities were more likely to elicit fast saccades than very low or very high numerosities. Reaction-times for vocal responses (collected in a separate experiment) were slower, did not depend on numerical range, and correlated only with the slow not the fast saccades, pointing to different systems. The short saccadic reaction-times we observe are surprising given that discrimination using numerosity estimation is thought to require a relatively complex neural circuit, with several relays of information through the parietal and prefrontal cortex. Our results suggest that fast numerosity-driven saccades may be generated on a single feed-forward pass of information recruiting a primitive system that cuts through the cortical hierarchy and rapidly transforms the numerosity information into a saccade command.

## 1. Introduction

The ability to rapidly estimate the number of enemies or prey, or food sources, can have obvious evolutionary benefits. Many animals, including primates [1], birds [2,3], fish [4], and even insects [5], can discriminate the number of elements in a scene, and many—including honeybees, even have the concept of zero [6,7]. It has been proposed that humans and animals share a 'number sense' that enables them to quickly perceive the number of objects in an image [8,9]. This idea opened a vast debate on how numerosity is sensed: directly through dedicated mechanisms [10], or indirectly through the combination of non-numerical properties of the array, such as density and area [11].

Accumulating evidence from both behavioural [12] and neuroimaging studies [13,14] in humans supports the first hypothesis, and further suggests that numerosity is more salient than many non-numerical properties. Six-month-old infants can reliably detect twofold changes in dot number, but need a fourfold change in area for comparable detection [15]. Stroop-like interference paradigms, where adult participants compared ensembles of dots varying along both numerical and non-numerical dimensions, show that numerosity is difficult to ignore during non-numerical judgements, whereas the reverse interference (non-numerical information biasing numerosity) was much weaker [16,17]. Other studies found that primates (both human and non-human) spontaneously orient choices in quantity discrimination tasks based on numerosity. When

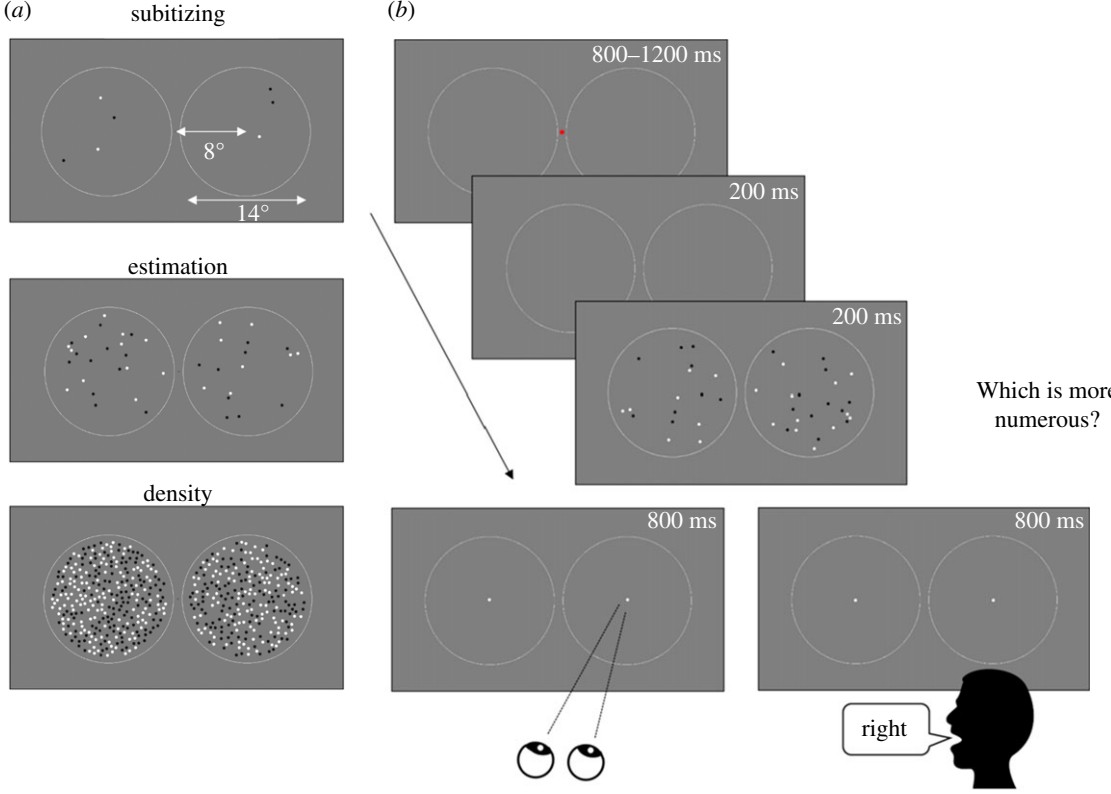

**Figure 1.** Stimuli and procedure. (*a*) Examples of stimuli targeting the subitizing, estimation, and density ranges. (*b*) Example of the time course of trials for saccadic and vocal reaction times. Participants maintained gaze on a central fixation point that disappeared after a pseudo-random interval (800–1200 ms). After 200 ms, two arrays of dots were briefly displayed. Participants either saccaded towards the most numerous array to one of the two landing points, or called out the side of the screen with the more numerous array.

identifying the odd-one-out of three dot arrays (without instructions on how the stimuli differ), human participants are far more sensitive to numerosity changes than to changes in area or density [18]. Similarly, using a categorization task in which arrays of dots could be labelled as 'little' or 'a lot', numerate adults and children, innumerate adults, and monkeys all based categorization on the numerical parameters rather than on other non-numerical dimensions [19]. Overall, these studies suggest that not only can numerical information be directly extracted from a visual scene, but it is the dimension to which we are most sensitive, and that most naturally attracts attention.

Many animals, particularly primates, constantly move their eyes to explore the surroundings, to monitor where they are heading and to direct gaze towards objects of interest. Saccadic eye movements can be extremely rapid, especially towards ecologically salient stimuli or possible threats [20,21]. Fischer & Boch [22] first described these *fast saccades* in monkeys, in response to the sudden appearance of a visual target against a homogeneous background. They found that saccadic onsets were distributed bimodally, with the first peak centred around 75 ms. In humans, similar paradigms triggered fast saccades with latencies of about 100 ms [23]. Interestingly, these fast saccades occur also for more complex stimuli if they are ecologically salient. When simultaneously presented with two images, one containing an animal or a human face, the other landscapes or vehicles, saccades towards the animal occurred within 120–130 ms [21,24], and towards faces within 100–110 ms [20]. It has been proposed that such ultra-rapid saccades might be achieved through hard-wired neural mechanisms developed under evolutionary pressure [20].

In light of the evidence that number is a highly salient visual dimension, we asked whether humans can choose

the most numerous array of items with fast saccadic eye movements. We also tested whether the saccade behaviour depends on the numerical range, given the evidence for different mechanisms covering different ranges [12].

## 2. Material and methods

Fourteen adults (six males, 29 ± 5 yo) with normal or corrected-to-normal vision participated in the saccade experiment; 11 of these also completed the vocal-response experiment (five males, 29 ± 5 yo).

Stimuli comprised arrays of dots of 0.35° diameter, half white and half black, on a mid-grey background, constrained within a 14° diameter circle. The number of dots was selected to target the three ranges of numerosity perception: 1–4 dots the subitizing range; 12, 17, 24, 35 dots the estimation range; 158, 195, 240, and 296 dots the texture-density range (figure 1*a*). In order to match task difficulty across the estimation and density ranges, each numerosity pair in a trial differed by multiples of the just noticeable difference (JND) (either 2, 4, or 6 JNDs), based on the sensitivity estimates on adult subjects reported by Anobile *et al.* [25]. Average root-mean squared (RMS) contrast ratios between stimuli pairs were 0.70, 0.75, and 0.84 for the subitizing, estimation, and density ranges. Participants fixated a red 0.35° diameter fixation point while two circles with 15° diameter located at 8° horizontal eccentricity delimited the region within which dot arrays were displayed (figure 1*b*). After a pseudo-random interval (800–1200 ms) the fixation dot disappeared and only the circles remained onscreen for 200 ms before stimuli were presented (facilitating fast saccadic eye movements [21]). Two arrays of dots were displayed for 200 ms, then immediately replaced by two landing points, which remained onscreen for 800 ms. In the first experiment participants were asked to make a

saccadic eye movement towards the most numerous array, as quickly and as accurately as possible. The second experiment was identical, except that participants called out the side containing more dots ('left' or 'right'), as quickly and as accurately as possible. Then a central fixation point appeared, and the program waited for a keypress to start the following trial. Pairs of stimuli designed to target one of the three ranges were randomly chosen from a total of 18 different conditions (six conditions per range, obtained from the combination of two out of four possible numbers), with the larger numerosity randomly left or right. Each side and condition were tested six times.

Participants completed three sessions, for a total of 648 trials, 216 trials for each range. For one subject one saccade session was discarded due to technical problems with the eye movement recording. Participants performed 10 practice trials prior to each experiment. Stimuli were generated under Matlab 9.6 using PsychToolbox routines [26] run by a Macintosh laptop (MacBook Pro, Apple) and presented on an external screen placed at 57 cm from the observer. Eye movements were recorded by an infrared eye tracker (EyeLink 1000), sampling eye position at 1000 Hz. Saccadic reaction-time was measured from stimulus onset. At the beginning of the experiment a standard calibration routine was run. Vocal responses were recorded by the experimenter who pressed the space bar as soon as the participant called out the response. Reaction-time was measured from the stimulus onset to the keypress.

Eye-movement traces were preprocessed to exclude trials where saccades started before stimuli onset, those with saccadic amplitudes shorter than 3 degrees, and those where participants initiated a saccade towards one side but then inverted direction to land on the other. To this aim we estimated the saccadic direction between 50 and 100 ms after the saccadic onset and checked whether this was changed at 200 ms. A total of 7% of trials from both the subitizing and estimation range and 10% of trials from the density range were discarded due to unsteady fixation or corrective saccades. Analysis of saccadic amplitudes across numerical ranges is reported in the electronic supplementary material. For the second experiment we discarded vocal reaction-times faster or slower than 3 standard deviations from the mean reaction-time, calculated separately for each subject and session. Less than 2% of the trials were discarded from each range.

Data were first analysed by merging individual data to form an 'aggregate participant'. The reaction-time distribution of each numerosity range was binned into 10 ms time bins and plotted to show the proportion of correct and incorrect responses in each bin. The multi-modality of the distributions was verified by applying Hartigan's dip test statistic [27,28] to the reaction-time distribution. For each numerosity range, we estimated the minimum saccadic reaction-time by searching for bins containing significantly more correct than incorrect responses using a binomial test with a criterion of $p < 0.05$, following the method of Crouzet et al. [20] and Kirchner & Thorpe [21]. The minimum reaction-time was defined by identifying the first of five consecutive bins that reached the criterion set by the binomial test. To further test which of the three ranges had the highest proportion of fastest responses we calculated the cumulative sum of the proportion of trials as a function of reaction-time.

To evaluate the impact of the speed–accuracy trade-off, and to take into account possible differences in task difficulty, we calculated the inverse-efficiency score [29] by dividing the reaction-time by response accuracy for each bin. To test which range elicited the fastest saccades, for each subject we fitted the saccadic reaction-time histograms after merging all ranges with a kernel smoothing function (using the Matlab function 'histfit' with kernel option). This fitting procedure revealed two clear peaks in most participants, very similar to the aggregate data. We identified the two highest peaks of the distribution and the minimum between them. The saccadic reaction-time corresponding to this minimum

point was chosen to separate fast from slow saccades. Reaction-times, accuracies, inverse-efficiency scores, and proportion of correct fast responses between ranges were entered into a repeated measure ANOVA (with three levels of numerical ranges). Bonferroni corrected post hoc comparisons and corresponding log 10 Bayes Factors are reported. By convention, base 10 logarithm of the Bayes Factor (logBF) > 0.5 is considered substantial evidence in favour of the alternative hypothesis, logBF > 1 strong evidence, and logBF > 2 decisive evidence. logBF < −0.5, −1, or −2 is substantial, strong, or decisive evidence in favour of the null hypothesis.

# 3. Results

## (a) Saccades

Participants saccaded to the more numerous of two briefly presented dot arrays. Figure 2a shows saccadic reaction-time histograms separately for correct and incorrect saccades, for each numerosity range. Following Kirchner & Thorpe [21], we estimated the minimum times required to initiate a correct saccade, adapting their method to the data pooled across participants. Saccades in the estimation range were the fastest, with minimum saccadic reaction-times of 190 ms. On the other hand, the minimum saccadic reaction-times in the subitizing and density ranges were 30–40 ms slower, respectively, 220 and 230 ms.

The distributions of correct saccades followed two distinct peaks, one fast (190–340 ms) and one slower (360 to approx. 600 ms). Bimodality was confirmed by Hartigan's dip test statistic, which was significant in all three ranges (all $p < 0.05$). On the basis of this division, we separated saccades into fast and slow subsets (greater or less than the value corresponding to the dip between the two peaks) and analysed them separately. The histograms of figure 2 clearly show that subjects were more likely to initiate fast saccades for stimuli in the estimation range than in the other two ranges. The blue curve in figure 2a shows that the highest proportion of correct saccades in the earliest time bins occurred in the estimation range. For a clearer visualization of the results we plotted the cumulative sum of the saccadic reaction-time distributions from the three ranges (figure 2b). The blue curve (estimation range) increased at a faster rate than the other curves, consistent with the higher proportion of fast saccades. To compensate for possible differences in task difficulty, we calculated the inverse-efficiency score by dividing saccadic reaction-times by response accuracy. Even after taking into account the speed–accuracy trade-off, the highest proportion of correct saccades initiated in the earliest time bins occurred when participants were tested with stimuli targeting the estimation range (figure 2c). This was observed also when plotting the cumulative sum of inverse-efficiency scores separately for the three ranges (figure 2d): the curve of the inverse-efficiency scores of the estimation range was much steeper than the other two.

The bimodality of the reaction-time histograms suggests that two different types of saccades occurred. To further study fast saccades, we selected those that were faster than the minima between the two peaks of the saccadic reaction-time distributions of all saccades, separately for each individual participant (see methods). The highest proportion was in the estimation range, reaching 38%, with only 29% and 19% in the subitizing and density ranges, respectively. ANOVA revealed a significant effect of range on the proportion of correct fast saccades ($F_{2,26} = 19.4$, $p < 0.001$). The proportion

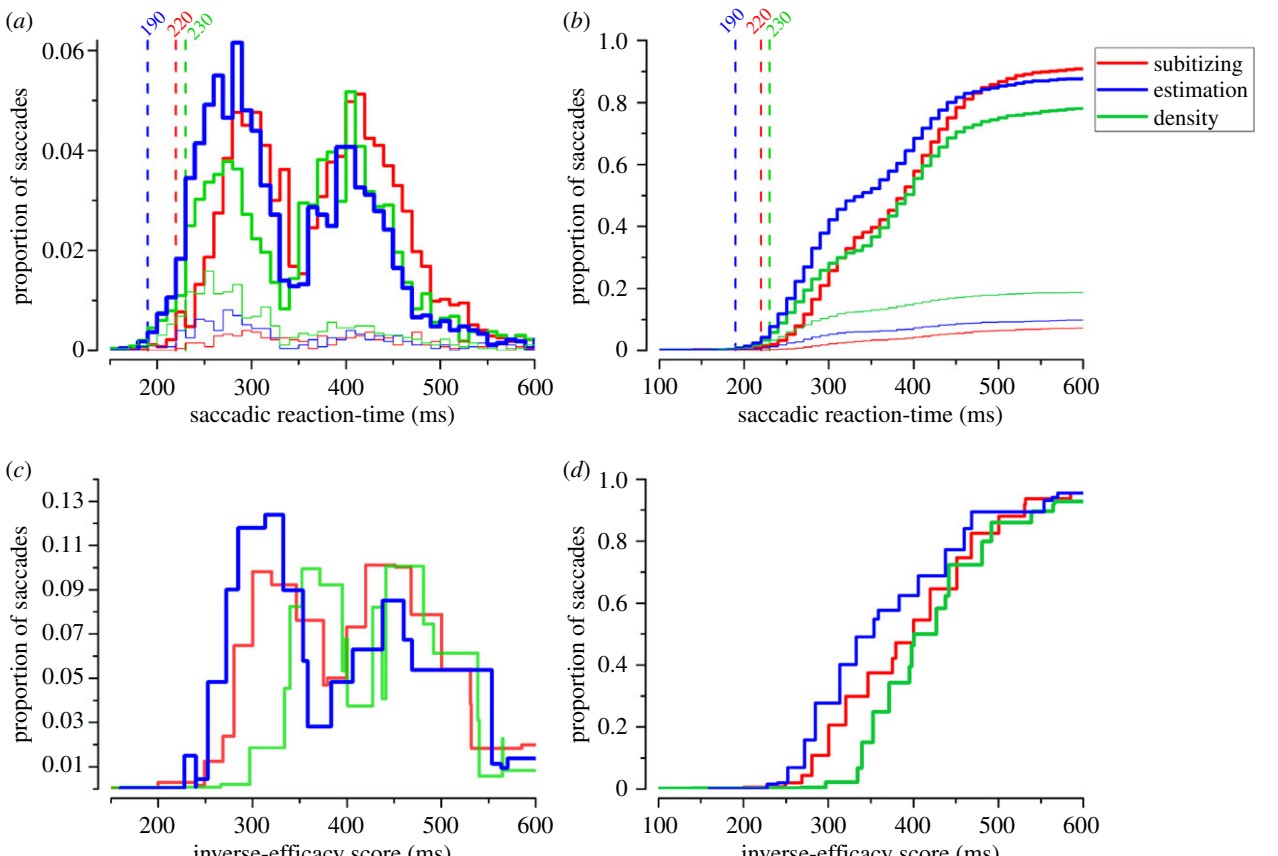

**Figure 2.** Saccadic reaction times. (*a*) Reaction-time histograms of correct (thick lines) and incorrect (thin lines) saccades for the subitizing (red), estimation (blue), and density (green) ranges. Dashed vertical lines refer to the minimum saccadic reaction-time for each range for reliably correct responses. (*b*) Cumulative sum of the proportion of saccades as a function of saccadic reaction-time. (*c*) Proportion of saccades as a function of the inverse-efficiency score (defined as the saccadic reaction-times divided by accuracy). (*d*) Cumulative sum of the proportion of saccades plotted as a function of the inverse-efficiency score. For visualization purposes 20-ms bins were used for the inverse-efficiency score plots.

in the estimation range was significantly higher than both subitizing ($t_{13} = 4.4$, $p = 0.002$, logBF = 1.7) and density ($t_{13} = 5.1$, $p < 0.001$, logBF = 2.2). Subitizing and density ranges also differed ($t_{13} = 3.1$, $p = 0.02$, logBF = 0.8).

We then tested whether saccades were on average faster for the estimation range, independently of whether the fast or slow saccades were selected. We quantified the average saccadic reaction-time for each participant, for each range. As shown in figure 3*a*, correct saccades were faster for stimuli in the estimation range (abscissa) than in the subitizing and density ranges (ordinate). This was confirmed by the significant effect of range in a repeated measures ANOVA ($F_{2,26} = 20.3$, $p < 0.001$). On average, saccades in the estimation range were performed in 338 ms, significantly faster than subitizing (373 ms; $t_{13} = 6.1$, $p < 0.001$, logBF = 2.1) and density (357 ms; $t_{13} = 4.9$, $p < 0.001$, logBF = 2.8) ranges. Saccadic reaction-times did not statistically differ between the subitizing and density ranges ($t_{13} = 2.4$, $p = 0.09$, logBF = 0.4). Importantly, these results were not explained by a difference in accuracy: although there was a significant difference in accuracy across ranges ($F_{2,26} = 26.7$, $p < 0.001$), this did not mirror the pattern of saccadic reaction-times (figure 3*b*). Saccadic accuracy in the density range (79%) was significantly lower than both subitizing ($t_{13} = 5.6$, $p < 0.001$, logBF = 2.5) and estimation ($t_{13} = 5.2$, $p < 0.001$, logBF = 2.3); saccadic accuracy in the estimation range (89%) was lower (although not significantly) than the subitizing range (92%, $t_{13} = 2.5$, $p = 0.08$, logBF = 0.41), inconsistent with the possibility that saccades in the estimation range were faster because the discrimination was easier.

As a more direct test to evaluate the impact of task difficulty on saccadic reaction-times, we compared inverse-efficiency scores between ranges (figure 3*c*). Inverse-efficiency in the estimation range was lower (386 ms) than that in the subitizing (410 ms) and density (455 ms) ranges. Repeated measures ANOVA revealed a significant effect of range ($F_{2,26} = 18.6$, $p < 0.001$), with inverse-efficiency for the estimation range significantly differing from both those for subitizing ($t_{13} = 3.01$, $p = 0.03$, logBF = 2.6) and for density ($t_{13} = 5.7$, $p < 0.001$, logBF = 0.7). Inverse-efficiency for density was significantly lower than subitizing ($t_{13} = 3.3$, $p = 0.02$, logBF = 0.9).

Distance effects, typical of magnitude judgements, occurred in all conditions, both when considering all saccades and only the fastest saccades: accuracy increased and reaction times decreased with larger numerical distances (see electronic supplementary material).

Overall, the results from this experiment showed that in general, the estimation range triggered faster saccades independently of accuracy, and that fast correct saccades are more likely to occur in this range than in the subitizing or density ranges.

## (b) Vocal responses

We repeated the experiment requiring participants to rapidly respond vocally, rather than move their eyes (figure 1*b*). With vocal rather than saccadic responses, reaction-time distributions for all ranges were unimodal (figure 4*a*: Hartigan's dip test statistic, all $p > 0.3$). Furthermore, the distributions

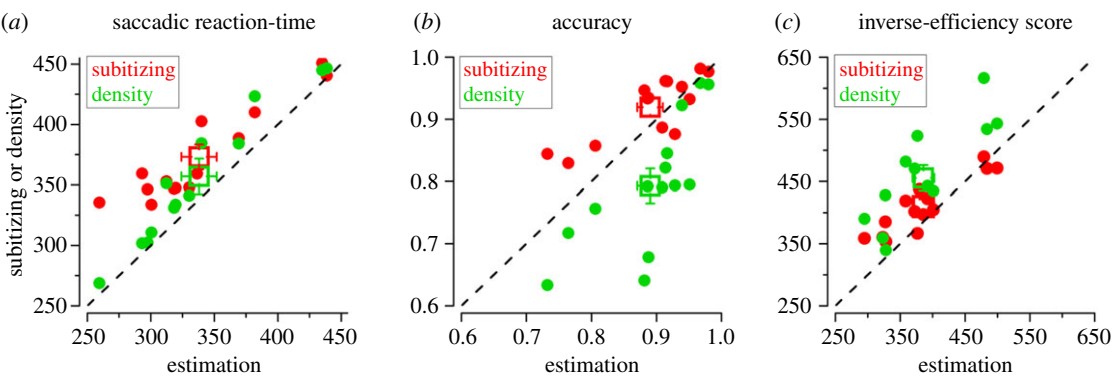

**Figure 3.** Individual results. Saccadic reaction-time for the correct responses (*a*, ms), accuracy (*b*, proportion correct), and inverse-efficiency score (*c*, ms) measured in the subitizing (red) and density (green) ranges (on the ordinate) plotted against those in the estimation range (on the abscissa). Individual participants are shown in circles, squares show the mean ± standard error of the mean ($n = 14$).

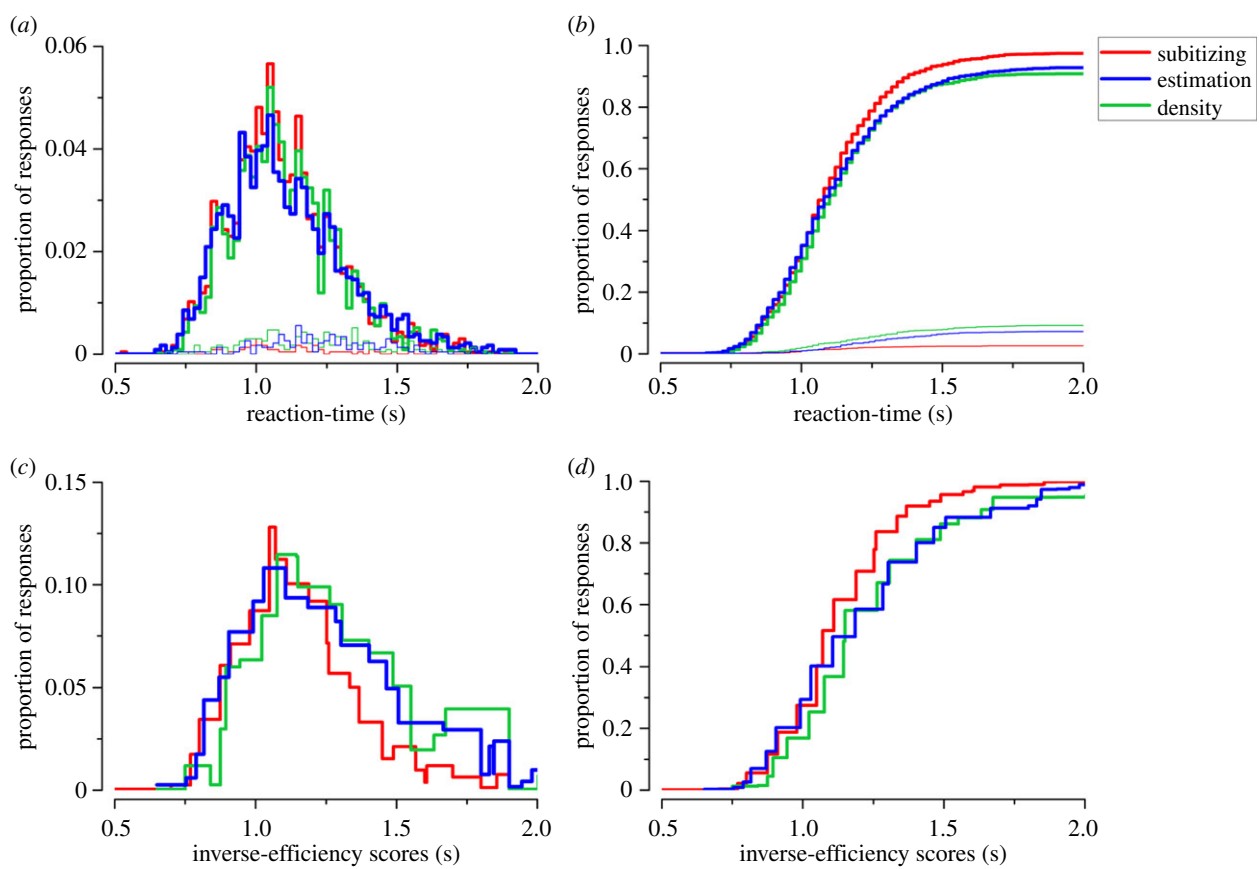

**Figure 4.** Vocal reaction-times. (*a*,*b*) Reaction-time histograms of correct (thick lines) and incorrect (thin lines) vocal responses in the subitizing (red), estimation (blue), and density (green) ranges. The distributions in the three ranges overlap, even when taking into account task difficulty by plotting the results as a function of the inverse-efficiency scores (*c*) and its cumulative sum (*d*). For visualization purposes 20-ms and 50-ms bins have been used for reaction-times and inverse-efficiency scores, respectively.

for the three numerosity ranges overlapped, with no clear advantage for the intermediate range (figure 4*c*,*d*). The reaction-time differences between ranges was quantified for individual participants. On average, participants gave the correct response in 1095 ms for the subitizing range and in 1102–1104 ms for the estimation and density ranges, not significantly different ($F_{2,20} = 0.44$, $p = 0.64$).

Response accuracies were statistically different between ranges ($F_{2,20} = 9.86$, $p = 0.001$), with the subitizing range significantly more accurate (97%) than the estimation (93%, $t_{10} = 3.8$, $p = 0.01$, logBF = 1.2) or density (91%, $t_{10} = 3.8$, $p = 0.01$, logBF = 1.2) ranges. Importantly, response accuracy did not significantly differ between the estimation and density ranges

($t_{10} = 1.3$, $p = 0.6$, logBF = −0.2), suggesting that task difficulty was successfully matched between these two ranges, at least when evaluated with vocal responses. Inverse-efficiency scores significantly differed between ranges ($F_{2,20} = 8.37$, $p = 0.002$), showing that when taking into account task difficulty, responses in the estimation and density ranges were significantly slower (1202 ms and 1236 ms, respectively) than those in the subitizing range (1127 ms, estimation versus subitizing: $t_{10} = 3.4$, $p = 0.02$, logBF = 0.9; density versus subitizing: $t_{10} = 3.7$, $p = 0.01$, logBF = 1.1). Inverse-efficiency scores did not statistically differ between the estimation and density ranges ($t_{10} = 1.2$, $p = 0.8$, logBF = −0.3). Vocal responses also showed distance effects, with accuracies increasing and reaction-times

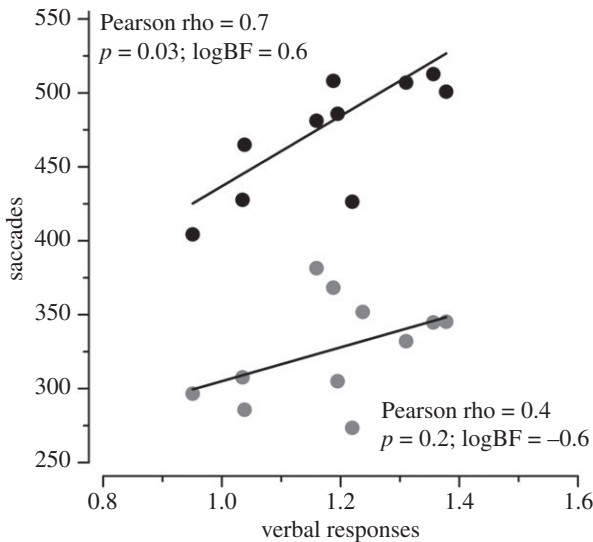

**Figure 5.** Correlational analysis. Correlation between the inverse-efficiency score calculated on the fast (grey dots) and slow (black dots) saccades and the inverse-efficiency score calculated on the vocal responses.

decreasing with larger numerical distances (see electronic supplementary material).

## (c) Relationship between saccades and vocal responses

We examined the relationship between vocal responses and saccades by correlating the inverse-efficiency scores of vocal responses against those of the fast and slow saccades (figure 5). Efficiency scores for vocal responses correlated significantly with those for slow saccades, with the Bayes Factor providing substantial evidence for the correlation ($r = 0.66$, $p = 0.03$, logBF = 0.6). However, fast saccades did not correlate with vocal reaction-times, with the Bayes Factor providing substantial evidence for lack of correlation ($r = 0.44$, $p = 0.17$, logBF = −0.6). The lack of correlation is further evidence that the fast saccades are driven by different circuitry than the vocal responses.

## 4. Discussion

In this study we show that participants can discriminate with saccadic movements the numerosity of briefly flashed dot ensembles as quickly as 190 ms. The bimodality of the saccadic onset distributions strongly resembled that observed in the studies that first described fast saccades in humans and monkeys [22,23]. Saccadic reaction-times were fastest when discriminating intermediate numerosities, with latencies as low as 190 ms, and about 40 ms slower for both very low and very high numerosities. The results could not be explained by a speed–accuracy trade-off, as the fast saccades are as accurate as the slow (which is not always the case [30]). Nor could they be explained by differences in saccadic amplitudes or relative RMS contrast ratios in the different ranges.

Saccadic reaction times vary over a large range, depending on features of the target stimulus (including chromaticity, cone contrast [31,32], size, and shape [33]), task timing (e.g. gap/overlap paradigms), and task (e.g. discrimination versus detection). For example, in a simple detection task, saccadic reaction times can be as slow as 300–320 ms when cone contrast is low [32]. Changing the task from simple detection to discrimination

adds about 100 ms per choice alternative [34], obviously more for more difficult than simple tasks [30]. On the other hand, minimum choice saccadic reaction-times towards faces (100–110 ms) [20] and animals (120–130) [21] are lower than the saccade latencies reported here. However, these studies required participants to detect salient stimuli, whereas here, participants discriminated numerosities. Saccades towards a face require detection of face-like characteristics in only one stimulus, while numerosity judgements are by definition relative, requiring processing and comparison of both stimuli. Given the range of saccadic latencies observed for various stimuli and tasks, the 190-ms reaction-times to intermediate numerosities are really quite fast as numerical choices between two alternatives.

Most similar to the current experiment, a previous study [35] has reported fast saccades towards Arabic digits (1–9) with a minimum reaction-time of 230 ms. Beyond the obvious major differences between symbolic and non-symbolic numbers, it is surprising that the saccadic reaction-times measured in this experiment were even faster than those directed towards overlearned (though language mediated) symbolic digits.

Our study shows that numerosity can be accurately processed at very high speeds, suggesting that numerosity discrimination is automatic. Importantly, the fastest reaction times for numerical processing were detected with saccadic eye movements, whereas vocal reactions times showed no tendency for a bimodal distribution or for differences across ranges. While both vocal responses and saccades showed typical distance effects, vocal responses correlated between participants only with slow, but not fast saccades, suggesting that two different systems (one fast and one slow) support numerosity discrimination.

Our results provide further evidence for dissociation between perceptual report and motor action [36]. For example, fast saccades are immune to motion-induced mislocalization of a flash [37] or a bar [38], while slow saccades (greater than 250 and greater than 130 ms, respectively) were fooled by the illusory effect. The amplitude of short-latency saccades (less than or equal to 140 ms) was also only slightly affected by size adaptation, compared to slower saccades [39]. These studies therefore suggest that visuo-motor control may access sensory feed-forward signals before conscious perception is reached through feedback connections.

An interesting aspect of fast saccadic eye movements is that they are not under full voluntary control: even when explicitly asked to saccade towards a neutral image (a vehicle), participants cannot avoid saccading towards the more salient image of a face [20]. This suggests that fast saccades towards salient stimuli tend to be 'mandatory', and to rely only marginally on attention. It would be interesting to test whether a change in task instructions also affect saccadic reaction-times and accuracies in the current paradigm. If humans have a natural preference to automatically shift gaze towards the more numerous ensemble, then asking participants to saccade towards the *less* numerous array may significantly slow down reaction-times or increase errors. Beyond the relative saliency that one numerosity may have over another, the fact that the fastest saccades observed here were more likely to occur in the estimation range lends support to the claim that this system does not tap strongly attentional resources. Attention has a different impact on numerosity perception, depending on the numerosity range. Although subitizing was initially considered a pre-attentive and parallel process [40], several

studies have shown that depriving visual attentional resources by double tasks [41–43], inattentional blindness [44], and attentional blink paradigms [45–47] has a detrimental effect on enumeration accuracy and discrimination thresholds for very small numerosities. Likewise, reaction-times and sensory thresholds for discriminating extremely high numerosities in the density range are elevated when participants have to respond to a visual distractor task first [42]. On the contrary, numerical discriminations in the estimation range are less affected by the deprivation of visual attentional resources [41–43]. A recent study has described a patient with an attentional deficit (simultagnosia) who is highly impaired in discriminating very small and very high numerosities, while thresholds for intermediate numerosities are similar to healthy controls, consistent with the notion that numerical comparisons in the estimation range can be performed with minimal reliance on attentional resources [48].

The current experiment reinforces evidence for three separate regimes for number perception, and suggests that mechanisms operating in the estimation range are more direct and automatic. Saccades to targets within the estimation range were overall faster, and a higher proportion of these could be considered 'fast saccades'. However, it is important to note that although there were more fast saccades in the estimation range, all three numerical ranges had bimodal reaction-time distributions, implicating fast and slow systems in all ranges. Whether this results from 'leakage' of the estimation system to the other two ranges, or whether all three ranges have fast and slow processes (in different proportion) is difficult to distinguish from the current experiment.

What can this pattern of result reveal about the underlying neural mechanisms driving fast saccades? One intriguing possibility is that the fastest saccades occur for numerosities in the estimation range because information in this range needs to be pooled over fewer and larger receptive fields compared with the density range. This would be consistent with a recent adaptation study suggesting that receptive fields in the estimation range are larger than those in the subitizing or density ranges [49]. That study suggested that numerosities in the estimation range may be coded by parietal neurons with large receptive field sizes (estimated to cover up to 12 degrees), whereas perception of higher numerosities may arise from low-level feature analysis, most likely carried out by neurons in the early visual areas with smaller receptive field sizes. The faster reaction to numerosities in the estimation range supports this possibility.

Another possibility to explain the short saccadic latencies is that numerical comparisons may be based on feed-forward signals, thought to support the ultra-rapid oculomotor responses by the early visual pathways [50]. Ultra-rapid oculomotor responses are initiated by the superior colliculus [51], a structure that is highly interconnected with the frontal eye field (FEF) and the posterior parietal cortex (PPC), all areas involved in saccadic planning and execution. There is evidence that the systems controlling saccadic eye movements and numerosity perception interact [52–54] and are at least in part controlled by overlapping areas in the parietal cortex [55,56].

Behavioural [29] and electroencephalography (EEG) studies [57,58] have suggested that numerical information may be processed by primitive, relatively direct pathways. For example, there are greater facilitatory effects for monocularly than dichoptically presented stimuli [29], suggesting

that numerical processing may start even before the monocular signals are fused, perhaps in the human subcortex. Event-related potential studies also point to the possibility of early and direct encoding of numerical quantities [57,58], with the effect of segregating stimuli into (a number of) perceptual units arising around 150 ms after stimulus onset [59]. This early numerosity signal may originate in V3/V3A—the first of many areas modulated by attention to number [14].

These results suggest that fast saccades towards numerical arrays may be supported by a visual cortical pathway that resolves numerical comparison tasks, either at the level of V3, or through direct connections to parietal and frontal cortices, which then converge in the superior colliculus within the same feed-forward wave. Interestingly, studies in monkeys have identified direct connections between the superior colliculus and V3 as well as between V3 and the caudal part of FEF [60] and to the posterior parietal areas [61], providing a potential physiological substrate. Single-cell recording studies in monkeys [62] estimated a conduction time of 30–35 ms between the retina and V1 and another 20–25 ms for the superior colliculus to elicit a saccade; everything that is in between is visual processing. If we consider that the corresponding latencies in humans are probably longer, it is likely that three to four synapses, potentially involving V3, PPC, and FEF, are sufficient to support fast saccades toward ensembles. It would be interesting to test these possibilities directly, taking advantage of the fact that our saccade paradigm can be readily adapted to non-human primates, and other laboratory animals.

In conclusion, we report very fast oculomotor responses towards non-symbolic numerosities in a numerical comparison task, suggesting that numerical information is a highly salient, relevant, and automatically coded visual dimension. The probability of triggering these fast saccades depends on the numerical range: they are more likely to occur when discriminating intermediate numerosities, consistent with observations showing that perceptual responses in that range are more automatic, relying less on attention. By operating on feed-forward signals processed by a very early visual pathway, a phylogenetically ancient system may drive fast saccades for efficient identification of numerosity.

Ethics. Experimental procedures were approved by the local ethic committee (Comitato Etico Pediatrico Regionale - Azienda Ospedaliero-Universitaria Meyer - Firenze), in accordance with Declaration of Helsinki. Written informed consent was signed by participants prior to the research.

Data accessibility. Data used for this publication have been deposited into the Dryad Digital Repository: https://doi.org/10.5061/dryad.7d7wm37s6 [63].

Authors' contributions. All authors contributed to the study concept and to the design. Testing and data collection were performed by E.C. E.C. and P.B. performed the data analysis. All authors contributed to the interpretation of results. E.C. drafted the manuscript and D.B., M.T., and P.B. provided critical revisions. All authors approved the final version of the manuscript for submission.

Competing interests. We declare we have no competing interests.

Funding. This research has received funding from the European Union's Horizon 2020 research and innovation program under the Marie Skłodowska-Curie (grant agreement no. 885672 – DYSC-EYE-7T), from the Accademia dei Lincei (fellowship 'G. Guelfi per le ricerche nel campo della biomedicina o della biologia 2019') and from the European Research Council (ERC) under the European Union's Horizon 2020 research and innovation programmes (grant agreement no. 801715 – PUPILTRAITS and no. 832813 – GenPercept).

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
