## [Reviewer comments · Proceedings of the Royal Society B: Biological Sciences]

Review History

RSPB-2020-1236.R0 (Original submission)

Review form: Reviewer 1

Recommendation

Major revision is needed (please make suggestions in comments)

Scientific importance: Is the manuscript an original and important contribution to its field?

Excellent

General interest: Is the paper of sufficient general interest?

Excellent

Quality of the paper: Is the overall quality of the paper suitable?

Excellent

Is the length of the paper justified?

Yes

Should the paper be seen by a specialist statistical reviewer?

Yes

Do you have any concerns about statistical analyses in this paper? If so, please specify them explicitly in your report.

Yes

It is a condition of publication that authors make their supporting data, code and materials available - either as supplementary material or hosted in an external repository. Please rate, if applicable, the supporting data on the following criteria.

Is it accessible?

Yes

Is it clear?

Yes

Is it adequate?

Yes

Do you have any ethical concerns with this paper?

No

Comments to the Author

The authors report that saccades to stimuli containing the higher numerosity are as fast as reactive saccades. That leaves only little time for processing of numerosity discrimination, providing evidence for the fascinating idea that number is sensed directly. The same was not true for density discrimination, providing further support for the number sense.

I have some concerns, I am sure the authors can deal with. The first is a matter of terminology which should be easy to repair. In the second point I ask about balanced stimulation in terms of contrast on both sides. I am sure the authors can address that properly. For the rest, the methods are sound and the results are interesting for a broad audience.

I am concerned with the usage of the term "fast saccades". The saccade types the authors are referring to in the introduction has been termed "express saccades". However, these saccades are defined by a latency of ~120 ms. Latencies of numerosity-driven saccades reach down to 190 ms. I understand the authors wish to state that this is still very quick given the visual task that must be completed. But still, it does sound odd to open up the category of "fast saccades" given that the average latency of reactive is 180-200 ms. The current saccades are surprisingly low because they are not reactive but expressing a perceptual choice. And would guess that the visual task adds duration to the average latency but it does not. I would advise the authors to explain that better. I am convinced that saccade researchers will find the term "fast saccades" for 190 ms latencies not fitting. And does the term really represent the finding best? The authors want to highlight that numerosity drives saccades directly, like a perceptual feature, not like a complex dimension that has to be reconstructed from low-level features. I suggest to change the title to: "Numerosity drives saccades directly in humans". Because that is what it does. Numerosity produces reactive-like saccades, not saccades reflecting choices with longer-processing. Please comment on this with comparisons of latencies in other tasks (I know the authors have done that partly, but I think that it is good to show at which position in the spectrum of saccade latencies numerosity-driven saccades are).

It is hard to believe that saccade reaction times that reflect discrimination of numerosities in the subutilizing range (≤ 4 objects) should be quicker than saccade reaction times for numerosities that contain more objects. Can the authors rule out that saccades might be driven by which side contains more stimulation in terms of contrast? That difference becomes more drastic in the medium range than in the subutilizing range. As I see it this factor is not controlled in the current setup.

The authors compare their data with the results from classical Thorpe experiments using faces. They argue that to classify a face seeing only part of it might suffice whereas to classify

numerosity one needs to see the entire image. I doubt that this interpretation is correct. If it would be relevant how much of an image is perceived why are latencies for subitizing stimuli not fastest where maximally 4 objects have to be registered. The easier explanation would be that Thorpe's task is a detection task whereas the current task is a discrimination task, notoriously involving more processing steps. That makes me wonder if the authors tried to make a Thorpe-like experiment with a numerosity stimulus either left or right. This task runs in the same problem as mentioned before: How to present a balanced stimulus on the opposite site that contains the same average contrast but no numerosity. Another option would be to test saccades after numerosity adaptation. That allows to present identical stimuli. If the authors should have no reply to my second point, that control might be an option.

Did the authors check whether there were significant differences in saccade amplitudes? Any difference might influence saccade latency. They should report saccade amplitudes.

Small numbers are processed by neurons with larger receptive fields than higher numbers (given the neural level of operation for both). For behavioral data this has been shown by Zimmermann, 2018, Cognition. Could this explain the higher processing speed (i.e. lower latency) for higher numbers? In other words, a density comparison must pool over information from many small receptive fields, a numerosity comparison over a little number of large receptive fields.

Minor fixes:

line 185-186: the side [that] contained more dots

line 107: "Participants fixated a red 0.35° fixation point" radius?

line 107: located at 8° of eccentricity horizontal!

Ref 21: They must also cite Zimmermann et al., 2016, Adaptation to size affects saccades with long but not short latencies.

Review form: Reviewer 2

Recommendation

Major revision is needed (please make suggestions in comments)

Scientific importance: Is the manuscript an original and important contribution to its field?

Excellent

General interest: Is the paper of sufficient general interest?

Excellent

Quality of the paper: Is the overall quality of the paper suitable?

Acceptable

Is the length of the paper justified?

Yes

Should the paper be seen by a specialist statistical reviewer?

No

Do you have any concerns about statistical analyses in this paper? If so, please specify them explicitly in your report.

No

It is a condition of publication that authors make their supporting data, code and materials available - either as supplementary material or hosted in an external repository. Please rate, if applicable, the supporting data on the following criteria.

Is it accessible?

No

Is it clear?

N/A

Is it adequate?

N/A

Do you have any ethical concerns with this paper?

No

Comments to the Author

This experiment addresses the question of whether numerosity processing can drive fast saccades in humans, much as faces or animals do in previous studies. In the saccade condition, a 200-ms display containing lateralized dot arrays was shown and participants needed to generate a saccade to the more numerous side of the screen. In the vocal condition, the same display occurred by participants needed to respond vocally left or right. Results show that correct saccades can occur extremely rapidly, and that this process may be most efficient in the estimation range versus the subitizing or density numeric ranges.

Main comments:

- Was there a minimum amplitude for a 1st saccade to be included in the analysis? What was the mean saccade amplitude and did it significantly differ between fast correct saccades and slow correct saccades across subjects?
- In the Methods, it says data were first analyzed by merging results across subjects, then later it says that to determine which range had the fastest saccades, individual subject analysis was done. Kirchner & Thorpe did the 10-ms bins and correct vs incorrect saccade analysis on individual participants. The results section suggests that the minimum correct RT was done as in that citation, but the methods suggest maybe this was done on merged data. This makes it unclear where the key saccadic latencies of 190, 220 and 230 ms come from.
- The fast and slow saccade distributions were separated using data merged across the range conditions. Visually, it would indeed seem that the two types of saccade distributions happened across all 3 numeric ranges. This seems to suggest that there is a common mechanism, and perhaps suggests it would be wrong to overemphasize the estimation range somehow is unique in this rapid processing.
- On p. 10, p. 13, and p.15, the authors talk about “typical distance effects” but there aren’t any statistics or details of these findings provided. Some pairs within range will have smaller absolute differences than others. If mean correct saccade latency (overall, fast, slow) is plotted versus the absolute magnitude of numerical difference between the two sides, would this show a linear change in saccade latency? Or would there be clear evidence of a different relationship between subitizing, estimating, and density judgements? A figure showing this would be helpful.
- On p. 14, the authors conclude that their results suggest numerosity discrimination is automatic and spontaneous. While they do provide some background literature on this claim in the introduction, the current experiments do not provide additional evidence of this, given that numerosity processing was a task-requirement in this experiment. Yes, saccadic responses were

shown to be fast, but it is unclear if they are automatic or what spontaneous might mean.

- The idea that “fast numerical processing was evident only with saccadic eye movements” seems also a questionable conclusion. After all, in the vocal condition, responses may have been slower, but accurate information was still clearly extracted rapidly from the brief 200-ms stimulus presentation. Other work has also show numerosity estimates can come from even briefer presentations and be reported in various ways, besides saccades. In the current paradigm, the vocal condition has the same 200-ms exposure to visual information, so it’s unclear how early processing versus later decision-making proceeded differently across tasks.

- In the Discussion on p.14 the authors note that “vocal responses to the same stimuli were much slower”. It seems hard to make conclusions about absolute reaction time from the current experiment, in which vocal RT was determined by the time at which an experimenter could make a key press in response to the participant’s vocal response. If participants were able to make button presses directly, RTs would be reduced considerably.

Minor:

- In terms of methods, I am unclear which pairs of numbers were used for each range. On p.6 it says there were 6 conditions per range, obtained from taking 4 numbers 2 at a time’. However at p.4-5, it says that number pairs displayed were chosen to be 2 JNDs apart. If each pair shown is exactly 2 JNDs apart, how are 6 conditions chosen from these 4 numbers? (It would appear left/right was a separate condition beyond these 6 from the text)

- Fig. 3a caption should note that what is plotted are saccadic latencies for correct trials only (assuming this is true).

Decision letter (RSPB-2020-1236.R0)

22-Jul-2020

Dear Professor Burr,

I am writing to inform you that we have now received referees' reports on your manuscript RSPB-2020-1236 entitled "Numerosity drives fast saccades in humans".

The manuscript has, in its current form, been rejected for publication in Proceedings B, for two reasons. Firstly, the paper does not make clear enough its value for addressing general biological principles, for example in providing insights into fundamental brain mechanism. Proc B is a general biology journal, and as such publications need to be accessible to a broad biological audience; my concern with the paper at present is that it feels better suited to a specialist psychology journal.. Secondly, whilst the specialist referees are generally positive about the paper's value, they have indicated that substantial revisions are necessary regarding several aspects. With this in mind we would be happy to consider a resubmission. However, you would need to ensure that provide a strong biological context to the work, and also fully address the comments of the referees. Please note that this is not a provisional acceptance, and if you feel these requirements are not feasible, I would suggest submitting to a more specialised journal.

The resubmission will be treated as a new manuscript, and we may approach new reviewers. Please note that resubmissions must be submitted within six months of the date of this email. In exceptional circumstances, extensions may be possible if agreed with the Editorial Office. Manuscripts submitted after this date will be automatically rejected.

Yours sincerely,
 Professor Loeske Kruuk
 mailto: proceedingsb@royalsociety.org

Editor comments: As per the comments above, please be sure to make the paper more accessible to a general biology audience. As a first suggestion, can you rephrase the title to something more informative for a broader readership?

Associate Editor
 Board Member: 1
 Comments to Author:

Both reviewers find the paper very interesting, but raise a number of significant concerns. Reviewer 1 questions the terminology used to describe the eye movements studied, and its relation to current usage in the field. They also have concerns about the most appropriate control conditions in order to rule out alternative explanations. Reviewer 2 raises detailed concerns about the way that data were analysed and tested statistically, and how the conclusions relate to the results.

Reviewer(s)' Comments to Author:
 Referee: 1

Comments to the Author(s)

The authors report that saccades to stimuli containing the higher numerosity are as fast as reactive saccades. That leaves only little time for processing of numerosity discrimination, providing evidence for the fascinating idea that number is sensed directly. The same was not true for density discrimination, providing further support for the number sense.

I have some concerns, I am sure the authors can deal with. The first is a matter of terminology which should be easy rot repair. In the second point I ask about balanced stimulation in terms of contrast on both sides. I am sure the authors can address that properly. For the rest, the methods are sound and the results are interesting for a broad audience.

I am concerned with the usage of the term "fast saccades". The saccade types the authors are referring to in the introduction has been termed "express saccades". However, these saccades are defined by a latency of ~120 ms. Latencies of numerosity-driven saccades reach down to 190 ms. I understand the authors wish to state that this is still very quick given the visual task that must be completed. But still, it does sound odd to open up the category of "fast saccades" given that the average latency of reactive is 180-200 ms. The current saccades are surprisingly low because they are not reactive but expressing a perceptual choice. And would guess that the visual task adds duration to the average latency but it does not. I would advise the authors to explain that better. I am convinced that saccade researchers will find the term "fast saccades" for 190 ms latencies not

fitting. And does the term really represent the finding best? The authors want to highlight that numerosity drives saccades directly, like a perceptual feature, not like a complex dimension that has to be reconstructed from low-level features. I suggest to change the title to: "Numerosity drives saccades directly in humans". Because that is what it does. Numerosity produces reactive-like saccades, not saccades reflecting choices with longer-processing. Please comment on this with comparisons of latencies in other tasks (I know the authors have done that partly, but I think that it is good to show at which position in the spectrum of saccade latencies numerosity-driven saccades are).

It is hard to believe that saccade reaction times that reflect discrimination of numerosities in the subutilizing range (≤ 4 objects) should be quicker than saccade reaction times for numerosities that contain more objects. Can the authors rule out that saccades might be driven by which side contains more stimulation in terms of contrast? That difference becomes more drastic in the medium range than in the subutilizing range. As I see it this factor is not controlled in the current setup.

The authors compare their data with the results from classical Thorpe experiments using faces. They argue that to classify a face seeing only part of it might suffice whereas to classify numerosity one needs to see the entire image. I doubt that this interpretation is correct. If it would be relevant how much of an image is perceived why are latencies for subutilizing stimuli not fastest where maximally 4 objects have to be registered. The easier explanation would be that Thorpe's task is a detection task whereas the current task is a discrimination task, notoriously involving more processing steps. That makes me wonder if the authors tried to make a Thorpe-like experiment with a numerosity stimulus either left or right. This task runs in the same problem as mentioned before: How to present a balanced stimulus on the opposite site that contains the same average contrast but no numerosity. Another option would be to test saccades after numerosity adaptation. That allows to present identical stimuli. If the authors should have no reply to my second point, that control might be an option.

Did the authors check whether there were significant differences in saccade amplitudes? Any difference might influence saccade latency. They should report saccade amplitudes.

Small numbers are processed by neurons with larger receptive fields than higher numbers (given the neural level of operation for both). For behavioral data this has been shown by Zimmermann, 2018, Cognition. Could this explain the higher processing speed (i.e. lower latency) for higher numbers? In other words, a density comparison must pool over information from many small receptive fields, a numerosity comparison over a little number of large receptive fields.

Minor fixes:

line 185-186: the side [that] contained more dots

line 107: "Participants fixated a red 0.35° fixation point" radius?

line 107: located at 8° of eccentricity horizontal!

Ref 21: They must also cite Zimmermann et al., 2016, Adaptation to size affects saccades with long but not short latencies.

Referee: 2

Comments to the Author(s)

This experiment addresses the question of whether numerosity processing can drive fast saccades in humans, much as faces or animals do in previous studies. In the saccade condition, a 200-ms display containing lateralized dot arrays was shown and participants needed to generate a saccade to the more numerous side of the screen. In the vocal condition, the same display

occurred by participants needed to respond vocally left or right. Results show that correct saccades can occur extremely rapidly, and that this process may be most efficient in the estimation range versus the subitizing or density numeric ranges.

Main comments:

- Was there a minimum amplitude for a 1st saccade to be included in the analysis? What was the mean saccade amplitude and did it significantly differ between fast correct saccades and slow correct saccades across subjects?

- In the Methods, it says data were first analyzed by merging results across subjects, then later it says that to determine which range had the fastest saccades, individual subject analysis was done. Kirchner & Thorpe did the 10-ms bins and correct vs incorrect saccade analysis on individual participants. The results section suggests that the minimum correct RT was done as in that citation, but the methods suggest maybe this was done on merged data. This makes it unclear where the key saccadic latencies of 190, 220 and 230 ms come from.

- The fast and slow saccade distributions were separated using data merged across the range conditions. Visually, it would indeed seem that the two types of saccade distributions happened across all 3 numeric ranges. This seems to suggest that there is a common mechanism, and perhaps suggests it would be wrong to overemphasize the estimation range somehow is unique in this rapid processing.

- On p. 10, p. 13, and p.15, the authors talk about "typical distance effects" but there aren't any statistics or details of these findings provided. Some pairs within range will have smaller absolute differences than others. If mean correct saccade latency (overall, fast, slow) is plotted versus the absolute magnitude of numerical difference between the two sides, would this show a linear change in saccade latency? Or would there be clear evidence of a different relationship between subitizing, estimating, and density judgements? A figure showing this would be helpful.

-On p. 14, the authors conclude that their results suggest numerosity discrimination is automatic and spontaneous. While they do provide some background literature on this claim in the introduction, the current experiments do not provide additional evidence of this, given that numerosity processing was a task-requirement in this experiment. Yes, saccadic responses were shown to be fast, but it is unclear if they are automatic or what spontaneous might mean.

- The idea that "fast numerical processing was evident only with saccadic eye movements" seems also a questionable conclusion. After all, in the vocal condition, responses may have been slower, but accurate information was still clearly extracted rapidly from the brief 200-ms stimulus presentation. Other work has also show numerosity estimates can come from even briefer presentations and be reported in various ways, besides saccades. In the current paradigm, the vocal condition has the same 200-ms exposure to visual information, so it's unclear how early processing versus later decision-making proceeded differently across tasks.

- In the Discussion on p.14 the authors note that "vocal responses to the same stimuli were much slower". It seems hard to make conclusions about absolute reaction time from the current experiment, in which vocal RT was determined by the time at which an experimenter could make a key press in response to the participant's vocal response. If participants were able to make button presses directly, RTs would be reduced considerably.

Minor:

- In terms of methods, I am unclear which pairs of numbers were used for each range. On p.6 it says there were 6 conditions per range, obtained from taking 4 numbers 2 at a time'. However at p.4-5, it says that number pairs displayed were chosen to be 2 JNDs apart. If each pair shown is

exactly 2 JNDs apart, how are 6 conditions chosen from these 4 numbers? (It would appear left/right was a separate condition beyond these 6 from the text)

- Fig. 3a caption should note that what is plotted are saccadic latencies for correct trials only (assuming this is true).

Author's Response to Decision Letter for (RSPB-2020-1236.R0)

See Appendix A.

RSPB-2020-1884.R0

Review form: Reviewer 1

Recommendation

Accept as is

Scientific importance: Is the manuscript an original and important contribution to its field?

Excellent

General interest: Is the paper of sufficient general interest?

Excellent

Quality of the paper: Is the overall quality of the paper suitable?

Excellent

Is the length of the paper justified?

Yes

Should the paper be seen by a specialist statistical reviewer?

No

Do you have any concerns about statistical analyses in this paper? If so, please specify them explicitly in your report.

No

It is a condition of publication that authors make their supporting data, code and materials available - either as supplementary material or hosted in an external repository. Please rate, if applicable, the supporting data on the following criteria.

Is it accessible?

Yes

Is it clear?

Yes

Is it adequate?

Yes

Do you have any ethical concerns with this paper?

No

Comments to the Author

First of all, I appreciate that the authors have responded to all my points in detail. I mentioned some potential confounds that the authors could rule out with extra analyses. My main point was the interpretation of "fast saccades". I argued that 190 ms might not be that fast. However, the authors provide a convincing estimate of the latencies with regard to other paradigms that supports their argument. I had already suggested in my first review that 190 ms leaves almost no time for the numerosity discrimination, which indeed is an extraordinary finding that makes the current study very important.

I think the study has wide implications for the biological sciences where numerosity perception is an active research topic in many species. I have no more comments to make and think that this is an excellent research report.

Review form: Reviewer 2

Recommendation

Accept with minor revision (please list in comments)

Scientific importance: Is the manuscript an original and important contribution to its field?

Good

General interest: Is the paper of sufficient general interest?

Good

Quality of the paper: Is the overall quality of the paper suitable?

Good

Is the length of the paper justified?

Yes

Should the paper be seen by a specialist statistical reviewer?

No

Do you have any concerns about statistical analyses in this paper? If so, please specify them explicitly in your report.

No

It is a condition of publication that authors make their supporting data, code and materials available - either as supplementary material or hosted in an external repository. Please rate, if applicable, the supporting data on the following criteria.

Is it accessible?

No

Is it clear?

N/A

Is it adequate?

N/A

Do you have any ethical concerns with this paper?

No

Comments to the Author

The new introduction that has been added is well written, and certainly increases the appropriateness for this journal in my opinion. Overall, the reviewers have addressed my comments, especially with the addition of the supplemental materials

As emphasized by the other reviewer, the fact that this is a discrimination task makes the findings even more impressive. I suggest an edit to line 32 of the abstract to say "given that discrimination using numerosity estimation is thought to require..."

There is still one place where the discussion of automaticity and its implications seem questionable. At the top of p.17, the authors cite a reference to a study that saccades to faces occur despite instruction otherwise, suggesting they are not under full voluntary control. This sentence should be followed by saying it is not clear how this maps onto numerosity and it is an empirical question as to whether participants could have followed instructions to saccade to the side with less items equally as well. (Additionally, the current sentence describing that citation does not seem to tie into well the rest of the paragraph focusing on less demand for attention in the estimation range specifically)

Decision letter (RSPB-2020-1884.R0)

01-Sep-2020

Dear Professor Burr

I am pleased to inform you that your manuscript RSPB-2020-1884 entitled "Fast saccadic eye-movements in humans suggest that numerosity perception is automatic and direct" has been accepted for publication in Proceedings B.

The referees and Associate Editor have recommended publication, but have also suggested some minor revisions to your manuscript. Therefore, I invite you to respond to the referee(s) comments and revise your manuscript. Because the schedule for publication is very tight, it is a condition of publication that you submit the revised version of your manuscript within 7 days. If you do not think you will be able to meet this date please let us know.

Sincerely,
Professor Loeske Kruuk
mailto: proceedingsb@royalsociety.org

Associate Editor

Comments to Author:

Both reviewers comment very favourably on the revised manuscript. One of the reviewers makes two further suggestions for minor changes to the text, which should be relatively straightforward to accommodate.

Reviewer(s)' Comments to Author:

Referee: 1

Comments to the Author(s).

First of all, I appreciate that the authors have responded to all my points in detail. I mentioned some potential confounds that the authors could rule out with extra analyses. My main point was the interpretation of "fast saccades". I argued that 190 ms might not be that fast. However, the authors provide a convincing estimate of the latencies with regard to other paradigms that supports their argument. I had already suggested in my first review that 190 ms leaves almost no time for the numerosity discrimination, which indeed is an extraordinary finding that makes the current study very important.

I think the study has wide implications for the biological sciences where numerosity perception is an active research topic in many species. I have no more comments to make and think that this is an excellent research report.

Referee: 2

Comments to the Author(s).

The new introduction that has been added is well written, and certainly increases the appropriateness for this journal in my opinion. Overall, the reviewers have addressed my comments, especially with the addition of the supplemental materials

As emphasized by the other reviewer, the fact that this is a discrimination task makes the findings even more impressive. I suggest an edit to line 32 of the abstract to say "given that discrimination using numerosity estimation is thought to require..."

There is still one place where the discussion of automaticity and its implications seem questionable. At the top of p.17, the authors cite a reference to a study that saccades to faces occur despite instruction otherwise, suggesting they are not under full voluntary control. This sentence should be followed by saying it is not clear how this maps onto numerosity and it is an empirical question as to whether participants could have followed instructions to saccade to the side with less items equally as well. (Additionally, the current sentence describing that citation does not seem to tie into well the rest of the paragraph focusing on less demand for attention in the estimation range specifically)

Author's Response to Decision Letter for (RSPB-2020-1884.R0)

See Appendix B.

Decision letter (RSPB-2020-1884.R1)

02-Sep-2020

Dear Professor Burr

I am pleased to inform you that your manuscript entitled "Fast saccadic eye-movements in humans suggest that numerosity perception is automatic and direct" has been accepted for publication in Proceedings B.

Your article has been estimated as being 9 pages long. Our Production Office will be able to confirm the exact length at proof stage.

Open Access

Paper charges

Sincerely,

Appendix A

We thank the editor and the reviewers for their thoughtful comments and questions, which we believe have contributed to a significant improvement of the manuscript. We hope that our replies and the associated changes in the manuscript (highlighted in blue in the marked copy) have addressed all their concerns and that this much improved paper is now suitable to be published.

Editor comments: As per the comments above, please be sure to make the paper more accessible to a general biology audience. As a first suggestion, can you rephrase the title to something more informative for a broader readership?

Reply: We do understand that *Proceedings B* is a general biology journal and that results should be generalizable to more than one specific species. There is considerable interest in numerosity perception across the entire animal kingdom, including non-human primates, birds, fish, and even insects. Recently there was much press about the finding that honeybees understand the concept of zero (Howard et al. Numerical ordering of zero in honeybees. *Science*, 2018). Books such as “the number sense” (Stanislas Dehaene) and “a brain for numbers” (Andreas Nieder) emphasise how numerical competency is deeply rooted in our biological ancestry. While our study is restricted to human observers, our approach provides a reliable physiological index that is widely used in other species, particularly non-human primates, based on fast eye movements. We therefore do believe that it is appropriate material for the *Proceedings*.

We have now recouched the motivation of our study to make it more interesting to the general biological reader, deferring much of the technical detail till later. We also point out that the eye-movement technology lends itself readily to pursuing with animal studies, particularly in primates. We have also changed the title to:

“Fast saccadic eye-movements in humans suggest that numerosity perception is automatic and direct”.

Referee: 1

The authors report that saccades to stimuli containing the higher numerosity are as fast as reactive saccades. That leaves only little time for processing of numerosity discrimination, providing evidence for the fascinating idea that number is sensed directly. The same was not true for density discrimination, providing further support for the number sense.

I have some concerns, I am sure the authors can deal with. The first is a matter of terminology which should be easy rot repair. In the second point I ask about balanced stimulation in terms of contrast on both sides. I am sure the authors can address that properly. For the rest, the methods are sound and the results are interesting for a broad audience.

Reply: we thank the reviewer for the positive feedback.

I am concerned with the usage of the term “fast saccades”. The saccade types the authors are referring to in the introduction has been termed “express saccades”. However, these saccades are defined by a latency of ~120 ms. Latencies of numerosity-driven saccades reach down to 190 ms. I understand the authors wish to state that this is still very quick given the visual task that must be completed. But still, it does sound odd to open up the category of “fast saccades” given that the average latency of reactive is 180-200 ms. The current saccades are surprisingly low because they are not reactive but expressing a perceptual choice. And would guess that the visual task adds duration to the average latency but it does not. I would advise the authors to explain that better. I am convinced that saccade researchers will find the term “fast saccades” for 190 ms latencies not fitting. And does the term really represent the finding best? The authors want to highlight that numerosity drives saccades directly, like a perceptual feature, not like a complex dimension that

has to be reconstructed from low-level features. I suggest to change the title to: “Numerosity drives saccades directly in humans”. Because that is what it does. Numerosity produces reactive-like saccades, not saccades reflecting choices with longer-processing. Please comment on this with comparisons of latencies in other tasks (I know the authors have done that partly, but I think that it is good to show at which position in the spectrum of saccade latencies numerosity-driven saccades are).

Reply: thanks for this comment. We agree with the reviewer that our latencies are longer than those for face detection. However, the saccades measured here are faster than those elicited by saccading towards stimuli defined by other basic visual features, such as colour, size, shape, etc (see the added paragraph for details). Crucially, as also noticed by the reviewer, the paradigm used here does not involve simple detection, but discrimination. We use the term “fast saccades” following Milosavljevic et al (2011) to refer to saccadic latencies of 230 ms elicited by a symbolic number comparison task, a paradigm comparable to ours. Therefore, given that we specify the exact latency of what we mean by ‘fast’ saccades in several points in the manuscript and that the term ‘fast saccades’ has been previously used for even longer latencies in the context of a discrimination paradigm most similar to ours, we prefer to maintain this terminology. We have expanded the discussion to include latencies of saccades in other tasks, and have also added a paragraph discussing the points the reviewer raised. (pages 14-15, line 347-366).

It is hard to believe that saccade reaction times that reflect discrimination of numerosities in the subitizing range (≤ 4 objects) should be quicker than saccade reaction times for numerosities that contain more objects. Can the authors rule out that saccades might be driven by which side contains more stimulation in terms of contrast? That difference becomes more drastic in the medium range than in the subitizing range. As I see it this factor is not controlled in the current setup.

Reply: Saccade reaction times that reflect discrimination of numerosities in the subitizing range (≤ 4 objects) were *slower* (presumably a typo) than saccade reaction times for numerosities that contain more objects, which is certainly surprising, a principle point for us, reflecting how estimation is relatively pre-attentive. Contrast is a good point. We now calculate the RMS contrast in each condition, and show that the ratios are least in the subitizing condition, higher for the medium range (as the reviewer anticipated) but highest for high densities, which does not predict the current results. This is now plotted in Supplementary Material.

The authors compare their data with the results from classical Thorpe experiments using faces. They argue that to classify a face seeing only part of it might suffice whereas to classify numerosity one needs to see the entire image. I doubt that this interpretation is correct. If it would be relevant how much of an image is perceived why are latencies for subitizing stimuli not fastest where maximally 4 objects have to be registered. The easier explanation would be that Thorpe’s task is a detection task whereas the current task is a discrimination task, notoriously involving more processing steps. That makes me wonder if the authors tried to make a Thorpe-like experiment with a numerosity stimulus either left or right. This task runs in the same problem as mentioned before: How to present a balanced stimulus on the opposite side that contains the same average contrast but no numerosity. Another option would be to test saccades after numerosity adaptation. That allows to present identical stimuli. If the authors should have no reply to my second point, that control might be an option.

Reply: thanks, we have deleted that interpretation and rephrased this sentence specifying that our paradigm differs from Thorpe’s. A “Thorpe-like” experiment would be ideal, but it is far from clear how to do that, as numerosity is not an entity in itself but *necessarily* involves a comparison between stimuli (or with a standard). Measuring saccades with adaptation is an excellent idea (thanks for suggesting it!), but altogether another experiment – as indicated in our reply to the reviewer’s second point, contrast or energy cannot explain our results given that contrast ratios increase monotonically across the three numerical ranges but saccade reaction times are fastest for the medium range (estimation) and slower both for the subitizing and the density ranges.

Did the authors check whether there were significant differences in saccade amplitudes? Any difference might influence saccade latency. They should report saccade amplitudes.

Reply: Thank you for raising this point. Amplitude differences cannot account for the latency differences, given that amplitudes monotonically decrease with numerosity while latency are the smallest for intermediate numerosities, as we now report in the Supplementary Material.

Small numbers are processed by neurons with larger receptive fields than higher numbers (given the neural level of operation for both). For behavioral data this has been shown by Zimmermann, 2018, Cognition. Could this explain the higher processing speed (i.e. lower latency) for higher numbers? In other words, a density comparison must pool over information from many small receptive fields, a numerosity comparison over a little number of large receptive fields.

Reply: thanks for this interesting comment which allowed us to speculate on the possible underlying neural mechanisms explaining the results. We have now added a paragraph to the discussion raising this interesting possibility, see pages 16-17 lines 409-419.

Minor fixes:

line 185-186: the side [that] contained more dots

line 107: "Participants fixated a red 0.35° fixation point" radius?

Reply: no, it was the diameter, we have added this detail in the text.

line 107: located at 8° of eccentricity horizontal!

Ref 21: They must also cite Zimmermann et al., 2016, Adaptation to size affects saccades with long but not short latencies.

Reply: thanks, these typos have been fixed and we are now citing Zimmermann et al., 2016 as well as Zimmermann et al. "Visual motion distorts visual and motor space" Journal of Vision (2012) 12(2):10, 1–8, see pages 15 line 375-378.

Referee: 2

Comments to the Author(s)

This experiment addresses the question of whether numerosity processing can drive fast saccades in humans, much as faces or animals do in previous studies. In the saccade condition, a 200-ms display containing lateralized dot arrays was shown and participants needed to generate a saccade to the more numerous side of the screen. In the vocal condition, the same display occurred by participants needed to respond vocally left or right. Results show that correct saccades can occur extremely rapidly, and that this process may be most efficient in the estimation range versus the subitizing or density numeric ranges.

Main comments:

- Was there a minimum amplitude for a 1st saccade to be included in the analysis? What was the mean saccade amplitude and did it significantly differ between fast correct saccades and slow correct saccades across subjects?

Reply: The minimum saccadic amplitude for a trial to be included in the analysis was 3 degrees, now specified in the method section. The Supplementary Material now reports the analyses on saccadic amplitudes. These tended to be smaller for faster saccades, but this was not seen across

numerical ranges (it was only reliable in the density range). In addition, amplitudes decrease monotonically with numerosity (whereas saccade reaction times are the fastest for the intermediate estimation range). These findings indicate that variations in saccadic amplitudes cannot explain the saccadic reaction time differences across numerical ranges.

- In the Methods, it says data were first analyzed by merging results across subjects, then later it says that to determine which range had the fastest saccades, individual subject analysis was done. Kirchner & Thorpe did the 10-ms bins and correct vs incorrect saccade analysis on individual participants. The results section suggests that the minimum correct RT was done as in that citation, but the methods suggest maybe this was done on merged data. This makes it unclear where the key saccadic latencies of 190, 220 and 230 ms come from.

Reply: Thanks for this advice, we now specify this:

“Following Kirchner & Thorpe², we estimated the minimum times required to initiate a correct saccade, adapting their method to the data pooled across participants.”

- The fast and slow saccade distributions were separated using data merged across the range conditions. Visually, it would indeed seem that the two types of saccade distributions happened across all 3 numeric ranges. This seems to suggest that there is a common mechanism, and perhaps suggests it would be wrong to overemphasize the estimation range somehow is unique in this rapid processing.

Reply: We agree and we thank the reviewer for raising this point, which we have incorporated in our discussion. In brief, we suggest that different numerical ranges preferentially tap into different mechanisms, but these are not exclusively reserved for any of these ranges.

We have added a paragraph to the discussion to address this issue (page 16, line 400-408).

- On p. 10, p. 13, and p.15, the authors talk about “typical distance effects” but there aren’t any statistics or details of these findings provided. Some pairs within range will have smaller absolute differences than others. If mean correct saccade latency (overall, fast, slow) is plotted versus the absolute magnitude of numerical difference between the two sides, would this show a linear change in saccade latency? Or would there be clear evidence of a different relationship between subitizing, estimating, and density judgements? A figure showing this would be helpful.

Reply: We left these analyses out of the original manuscript to keep within the length limits, but now include them in the Supplementary Material. We show distance effects, both in accuracy and reaction times, both for the saccade task (considering all saccades or selectively the fastest ones), and for the vocal response task.

-On p. 14, the authors conclude that their results suggest numerosity discrimination is automatic and spontaneous. While they do provide some background literature on this claim in the introduction, the current experiments do not provide additional evidence of this, given that numerosity processing was a task-requirement in this experiment. Yes, saccadic responses were shown to be fast, but it is unclear if they are automatic or what spontaneous might mean.

Reply: We agree with the reviewer that “automatic” is more appropriate than “spontaneous” to describe the current results, given that the task was numerosity discrimination and we edited the manuscript accordingly.

- The idea that “fast numerical processing was evident only with saccadic eye movements” seems also a questionable conclusion. After all, in the vocal condition, responses may have been slower, but accurate information was still clearly extracted rapidly from the brief 200-ms stimulus presentation. Other work has also show numerosity estimates can come from even briefer presentations and be reported in various ways, besides saccades. In the current paradigm, the vocal condition has the same 200-ms exposure to visual information, so it’s unclear how early processing versus later decision-making proceeded differently across tasks.

Reply: rephrased as follows: “Importantly, the fastest reaction times for numerical processing were detected with saccadic eye movements, whereas vocal reactions times showed no tendency for a bimodal distribution or for differences across the ranges.”

- In the Discussion on p.14 the authors note that “vocal responses to the same stimuli were much slower”. It seems hard to make conclusions about absolute reaction time from the current experiment, in which vocal RT was determined by the time at which an experimenter could make a key press in response to the participant’s vocal response. If participants were able to make button presses directly, RTs would be reduced considerably.

Reply: Appropriate button-press would certainly have been a valid alternative, but we chose the vocal estimation method instead. The experimenter maintained the finger on the button and stopped the time recording as soon as they heard the response. We found this more reliable than using voice-activated technologies, and found that our average reaction times for the estimation range (around 1100 ms) were comparable to those reported in the experiment by Pomé et al 2019 (around 1300 ms), with voice-activated technology. However, we were not particularly interested in absolute latencies, only in comparing between numerosity ranges. We assume that any extra latencies in this technique will occur for all numerosities.

Minor:

- In terms of methods, I am unclear which pairs of numbers were used for each range. On p.6 it says there were 6 conditions per range, obtained from taking 4 numbers 2 at a time¹. However at p.4-5, it says that number pairs displayed were chosen to be 2 JNDs apart. If each pair shown is exactly 2 JNDs apart, how are 6 conditions chosen from these 4 numbers? (It would appear left/right was a separate condition beyond these 6 from the text)

Reply: thanks for pointing this out; rephrased as “each numerosity pair in a trial differed by multiples of the JND (either 2, 4 or 6 JNDs)”

- Fig. 3a caption should note that what is plotted are saccadic latencies for correct trials only (assuming this is true).

Reply: added, thank you.

Appendix B

We again thank the editor and the reviewers for their work on this manuscript, which has brought significant improvements. We have incorporated these final suggestions in the manuscript, (highlighted in blue on pages 2 and 16 of the marked copy), and trust that it is now ready for publication.